# MYL9 deficiency is neonatal lethal in mice due to abnormalities in the lung and the muscularis propria of the bladder and intestine

Chu-Han Huang[1,2], Joyce Schuring[1,3], Jarrod P. Skinner[1], Lawrence Mok[1,2], Mark M. W. Chong[1,2]*

**1** St Vincent's Institute of Medical Research, Fitzroy, VIC, Australia, **2** Department of Medicine (St Vincent's), University of Melbourne, Fitzroy, VIC, Australia, **3** HAN University of Applied Sciences, Nijmegen, The Netherlands

* mchong@svi.edu.au

## Abstract

Class II myosin complexes are responsible for muscle contraction as well as other non-sarcomeric contractile functions in cells. Myosin heavy chain molecules form the core of these structures, while light chain molecules regulate their stability and function. MYL9 is a light chain isoform that is thought to regulate non-sarcomeric myosin. However, whether this in only in specific cell types or in all cells remains unclear. To address this, we generated MYL9 deficient mice. These mice die soon after birth with abnormalities in multiple organs. All mice exhibited a distended bladder, shortening of the small intestine and alveolar over-distension in the lung. The *Myl9* allele in these mice included a LacZ reporter knockin that allowed for mapping of *Myl9* gene expression. Using this reporter, we show that MYL9 expression is restricted to the muscularis propria of the small intestine and bladder, as well as in the smooth muscle layer of the bronchi in the lung and major bladder vessels in all organs. This suggests that MYL9 is important for the function of smooth muscle cells in these organs. Smooth muscle dysfunction is therefore likely to be the cause of the abnormalities observed in the intestine, bladder and lung of MYL9 deficient mice and the resulting neonatal lethality.

## Introduction

Distinct class II myosin complexes, also known as conventional myosin, are found in muscle cells, where they are responsible for muscle contraction, and in all cells, where they are required for the contractile bundles involved in cellular structure, motility, adhesion and cytokinesis. Myosin II complexes are composed of two myosin heavy chains (MHCs) and four myosin light chains (MLCs). The MHCs are responsible for oligomerization and interactions with actin, while the MLCs are required for maintaining the stability of the holocomplex [1]. MLCs also regulate the mechanoenzymatic activities of the complex. MLCs are divided into

**Funding:** This work was supported by grants and fellowships from the National Health and Medical Research Council, Australia (www.nhmrc.gov.au 1079586, 1117154, 1122384 and 1122395 to MMWC), Diabetes Australia (www.diabetesaustralia.com.au Y20G-CHOM to MMWC), U.S. Department of Defense (www.cdmrp.army.mil W81XWH-19-1-0728 to MMWC) and the Perpetual Foundation - The Ann Helene Toakley Charitable Endowment (www.perpetual.com.au to MMWC). This work was made possible through Victorian State Government Operational Infrastructure Support and Australian National Health and Medical Research Council Research Institute Infrastructure Support Schemes. The funders had no role in study design, data collection and analysis, decision to publish, or preparation of the manuscript.

**Competing interests:** The authors have declared that no competing interests exist

two subgroups, regulatory light chains or essential light chains, based on whether they primarily regulate the function or the structural integrity of the myosin complex.

The function of myosin II complexes in different cell types is dependent on specific MLCs that interact with the MHCs. Myosin light chain 1 (MYL1) is essential for skeletal muscle function [2, 3], while MYL2 and MYL3 are important in ventricular cardiac muscle and MYL4 is required in atrial cardiac muscle [4, 5].

Three closely related MLCs, MYL12A, MYL12B and MYL9 are thought to regulate non-sarcomeric myosin. MYL12A shares 99% sequence homology with MYL12B and 96% with MYL9 [6]. These three MLCs appear to preferentially interact with MHCs of non-muscle cells, although they have also been shown to interact with muscle MHCs [6, 7]. Knockdown of MYL12A or MYL12B disrupts cell morphology due to destabilization of myosin II complexes [6].

Whether MYL9 is also required for non-muscle myosin stability isn't clear but it has been shown to exert a range of non-muscle functions. MYL9, as well as MYL12A and MYL12B, has been found in T cells where it has been shown to have a role in positioning CD3 to the cell surface [8]. CD3 is a critical protein complex required for transducing signals from the T cell receptor. MYL9 also expressed in megakaryocytes and knockdown impairs platelet production [9]. Interestingly, the MYL9, along with MYL12A/B, derived from platelets has been shown to function as a ligand for CD69 expressed by activated T cells in allergic airway inflammation [10]. Knockdown of MYL9 has also been shown to impair endothelial cell migration *in vitro* [11]. Upregulation of MYL9 has been observed in several cancers, including colorectal cancer and glioblastoma, where it is thought to promote the proliferation and migration of the cancer cells [12, 13].

We previously showed that DROSHA deficiency impairs the pluripotency of hematopoietic stem cells and this is due to the aberrant upregulation of MYL9 [14]. DROSHA is an RNase III enzyme best known for its role in the microRNA biogenesis pathway [15]. However, in stem cells, it also functions to degrade specific messenger (m)RNA targets, which occurs independently of microRNAs [16]. In hematopoietic stem cells, DROSHA recognizes a secondary stem-loop structure in the *Myl9* mRNA and this normally results in RNA degradation and suppression of MYL9 expression [14]. Why the *Myl9* gene is actively transcribed in hematopoietic stem cells and then suppressed by this RNA degradation mechanism is unknown.

While MYL9 has been associated with a wide range of pathologic conditions, its normal physiologic function remains poorly understood. RNA profiling previously showed that it has a wide distribution, with at least low-level expression detected in all tissues [6]. However, whether MYL9 is expressed in all cells or is restricted to non-muscle cells remains contentious due to the lack of antibodies that can specifically detect MYL9. This is because the >96% amino acid homology with MYL12A and MYL12B results in significant cross-reactivity of antibodies. In this study, we investigate the expression and function of MYL9 by generating a LacZ-knockin/knockout mouse model. We investigate the impact of MYL9 deficiency and employ the LacZ-reporter gene to map the expression of MYL9 within tissues and during embryonic development.

## Materials and methods

### Mice

Three embryonic stem cell clones, designated Myl9$^{\text{tm1a(KOMP)Wtsi}}$, were obtained from the Mutant Mouse Resource & Research Centre at the University of California, Davis. The clones, on the C57BL/6 background, contained a heterozygous knockin of a LacZ–Neomycin resistance cassette that disrupted the *Myl9* gene. All three clones were microinjected into

blastocysts at Phenomics Australia (Monash Node) to produce chimeras. Germline transmission was achieved from all three clones.

Time-pregnancies of female mice were achieved by placing with a male in the afternoon. The female was then check for a vaginal plug the next morning. If present, this was designed day E0.5 of gestation. Embryos were then harvested at specific times during gestion or postpartum.

Genotyping of animals was performed by PCR with the primers: 5'-CGTCTCCCAGAT GTGCAGTA and 5'-CTCTGACACACACCCACCAC, which detects both the wildtype allele (200bp product) and knockin/out allele (260 product).

All mice were housed at St Vincent's Hospital Melbourne's BioResources Centre. Experiments were approved by the Animal Ethics Committee of St Vincent's Hospital Melbourne and performed under the Australian code for the care and use of animals for scientific purposes. Adult mice were euthanized by asphyxiation with $CO_2$. Newborn pups were euthanized by decapitation.

## Quantitative RT-PCR

Total RNA was extracted from tissues with TRIsure (Meridian Bioscience, Cincinnati) and reverse transcribed with M-MuLV Reverse Transcriptase (New England BioLabs, Ipswich). Quantitative PCR was then performed using GoTaq qPCR Master Mix (Promega, Madison) with the primers: 5'-ATGAGGAGGTGGACGAGATG and 5'-CACGGGGAGGGTAGAGTGTA to detect exon 4 in wildtype *Myl9* or 5'-ATCACGACGCGCTGTATC and 5'-ACATCGGGC AAATAATATCG to detect LacZ.

## Analysis of the Genotype-Tissue Expression (GTEx) database

The GTEx Project was supported by the Common Fund of the Office of the Director of the National Institutes of Health, and by NCI, NHGRI, NHLBI, NIDA, NIMH, and NINDS. Expression data for *MYL9*, *MYl2A* and *MYL12B* were obtained from the GTEx Portal on 01/03/22.

## Whole embryo X-gal staining

Embryos were harvested at gestational stage E14.5, rinsed with PBS and fixed in 4% paraformaldehyde in PBS for 2h on ice. They were then incubated with 5-bromo-4-chloro-3-indolyl-β-D-galactopyranoside (X-gal) (Sigma-Aldrich, St Louis) as previously described [17] to detect β-galactosidase activity. Following X-gal staining, the embryos were cleared by incubating in 1% KOH for 30min, followed by 1% KOH / 20% glycerol for 1h, then 1% KOH / 33% glycerol for 4h and finally overnight in 1% KOH / 50% glycerol. The embryos were stored in 2% PFA / 50% glycerol until imaging.

## X-gal and immunohistochemical staining of frozen sections

Organs were collected from newborn pups at D0 postpartum or from 6 week old adult mice. The organs were rinsed with PBS and fixed in 4% paraformaldehyde for 4-6h. This was followed by two washes in PBS. The organs were then infused overnight with 30% sucrose in PBS and finally frozen in Tissue-Tek O.C.T. Compound (Sakura Finetek, Torrance). Following sectioning on a cryostat, frozen sections were stained with hematoxylin and eosin or subjected to immunohistochemical and/or X-gal staining.

To detect β-galactosidase activity, the sections were first stained with X-gal overnight as previously described [18]. They were then blocked with 0.5% BSA (Sigma-Aldrich) in PBS for

30min before immunohistochemical staining for α-Smooth Muscle Actin (α-SMA) with a mouse anti-α-SMA monoclonal antibody (clone 1A4, Thermo Fisher Scientific, Waltham) followed by a horse radish peroxidase-conjugated horse anti-mouse secondary antibody (Cell Signaling Technology, Danvers). The anti-α-SMA staining was then developed with the Pierce DAB Substrate Kit according to the manufacturer's instructions (Thermo Fisher Scientific). Finally, the sections were counterstained with Nuclear Fast Red (Sigma-Aldrich), dehydrated and mounted.

Immunohistochemical staining of sections for MYL9 protein was performed with a rabbit anti-MYL9/12A/12B monoclonal antibody (clone EPR13013[2][B], Abcam, Cambridge, UK) followed by a horse radish peroxidase-conjugated goat anti-rabbit secondary antibody (Cell Signaling Technology) and with the Pierce DAB Substrate Kit. The sections were counterstained with Nuclear Fast Red (Sigma-Aldrich), dehydrated and mounted.

### Fluorescence-activated cell sorting (FACS)-gal analysis

Detection of β-galactosidase activity in hematopoietic cells was performed essentially as described previously [17]. Briefly, single cell suspensions in PBS + 5% FCS were loaded hypotonically by diluting 1:1 with 2mM fluorescein di-β-D-galactopyranoside (FDG) (Sigma-Aldrich) and incubating at 37˚C for 2min. Loading was stopped by adding 10× cold PBS + 5% FCS and the cells were incubated on ice for 3h. The cells were than stained with various cell surface antibodies to identify different hematopoietic populations. The following anti-mouse antibodies (all from Thermo Fisher Scientific, Waltham) were employed: TCRβ (clone H57-597), TCRγδ (clone eBioGL3), B220 (clone RA3-6B2), Sca1 (clone D7), cKit (clone 2B8), CD34 (clone RAM34), CD16/32 (clone 93), CD127 (clone eBioSB/199) and CD135 (clone A2F10). A lineage antibody cocktail (Miltenyi Biotec, Bergisch Gladbach, Germany) used to identify mature bone marrow cells. The labelled cells were then analyzed on an LSRFortessa (BD Biosciences, Franklin Lakes). Subsequent data analysis was performed on FlowJo ver10.7 (BD Biosciences).

### Statistical analyses

All statistical analyses were performed with Prism ver9 (GraphPad, San Diego). Comparisons between $Myl9^{+/+}$ and $Myl9^{-/-}$ pups were assessed by $t$-test.

## Results

### Neonatal lethality in MYL9 deficient mice

To better understand the physiologic function of MYL9, we generated mice with MYL9 deficiency. Three embryonic stem (ES) cell clones with a *LacZ*–Neomycin resistance cassette knocked into the *Myl9* gene between exons 2 and 3 were obtained from the Knockout Mouse Repository, USA (Fig 1A). The *LacZ* cassette contains a splice acceptor and polyA tail, which results in a fusion transcript with exons 1 and 2 of the *Myl9* gene. This excludes exons 3 and 4 of the *Myl9* gene and produces β-galactosidase in place of MYL9 protein. This *Myl9* null allele therefore also functions as a reporter for *Myl9* promoter activity. The ESC cells were injected into blastocysts to produce chimeras. Germline transmission was obtained for all three ES cell clones. Two lines, 576 and 577, were then selected for further analysis.

Heterozygous $Myl9^{-/+}$ male and female mice were mated together to generate $Myl9^{-/-}$ offspring, with the 576 and 577 lines maintained separately. We initially screened for genotypes at weaning but was unable to identify any $Myl9^{-/-}$ animals in either line. To determine when the $Myl9^{-/-}$ animals were being lost, litters were analyzed at different stages during embryonic

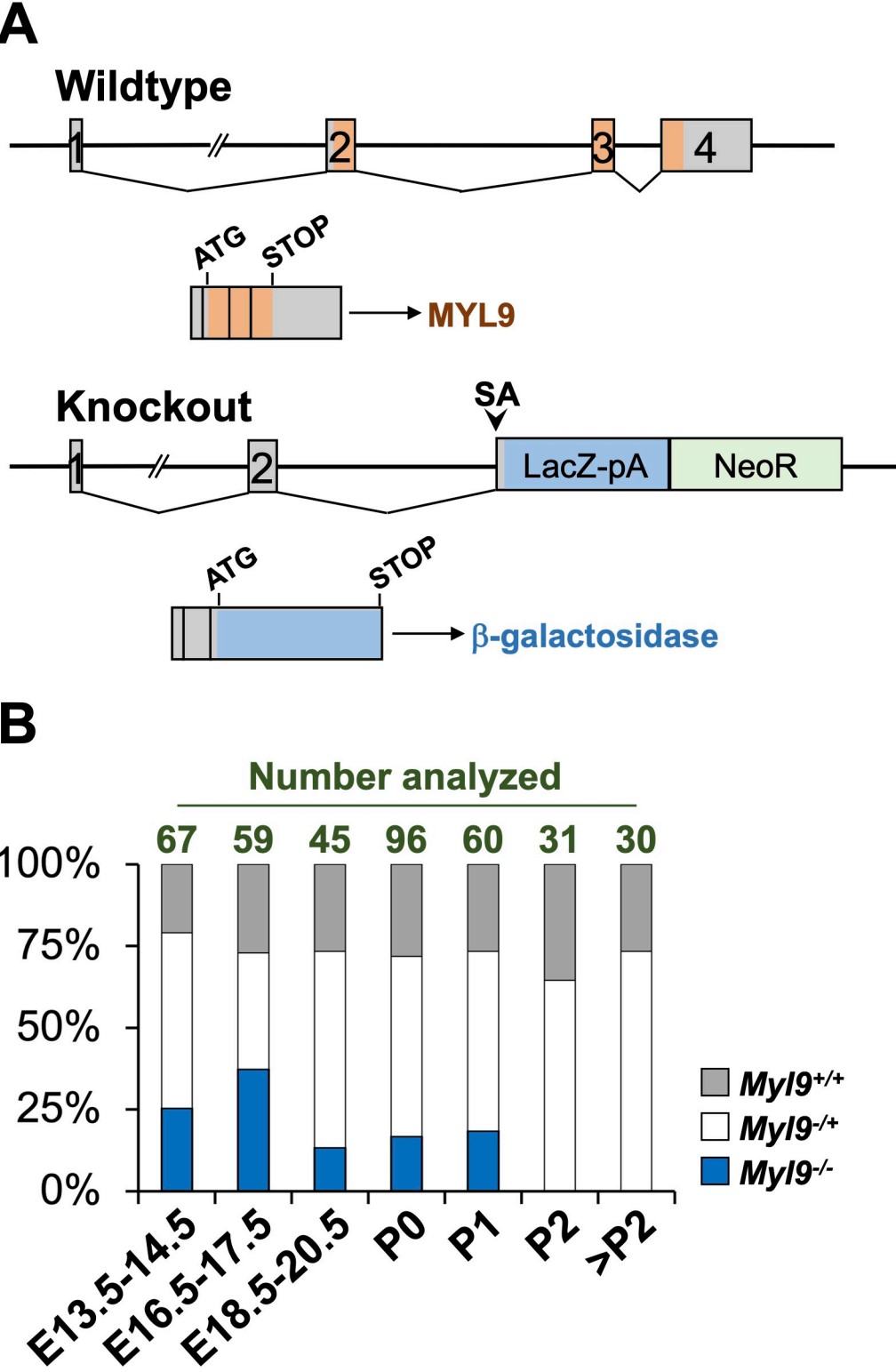

**Fig 1. MYL9 deficiency is cause neonatal lethality in mice. (A)** MYL9 deficient mice were generated by inserting a LacZ-Neomycin resistance cassette between exons 2 and 3 of the *Myl9* gene. The LacZ component includes a splice acceptor (SA) and polyadenylation signal (pA), while allows its incorporation into the transcript in place of exons 3 and 4 of the *Myl9* gene. Detection of β-galactosidase functions as a reporter for *Myl9* promoter activity. **(B)** Heterozygous *Myl9*⁻ᐟ⁺ female mice were time-mated with heterozygous *Myl9*⁻ᐟ⁺ male mice, then embryos or newborn pups were

genotyped at the indicated time-points. Shown are the percentages of wildtype *Myl9*+/+, heterozygous *Myl9*-/+ and homozygous *Myl9*-/- offspring obtained at each timepoint. The number of embryos/pups analyzed at each timepoint is indicated above the bars. The individual data points are provided in S1 File.

development or soon after birth (Fig 1B). Both the 576 and 577 lines exhibited the same survival, and the data shown combines the embryos/pups from both lines. Mendelian ratios were roughly normal at all embryonic stages analyzed, with *Myl9*-/- embryos observed at E13.5 through to E18.5~E20.5. *Myl9*-/- pups were still observed at postpartum Day 0 and 1. However, no *Myl9*-/- pups were observed at Day 2 after birth. Therefore, Myl9 deficiency is neonatal lethal in mice soon after birth. As both lines exhibited the same survival, all subsequent phenotypic characterizations were performed only in the 577 line.

## MYL9 is expressed in the developing lung, intestine and urinary tissues

The neonatal lethality caused by MYL9 deficiency suggests that the protein must be important for the development and/or function of a critical physiologic system. To determine where MYL9 functions and therefore the cause of this lethality, we analyzed *Myl9*-/+ embryos for β-galactosidase activity as a reporter for MYL9 expression. *Myl9*-/+ female mice were mated with wildtype *Myl9*+/+ male mice and embryos were harvested at E14.5 in development for analysis by whole embryo X-gal staining (Fig 2). Half the embryos were *Myl9*-/+ and expressed β-galactosidase from one allele of *Myl9*, while the other half of embryos were *Myl9*+/+ and served as negative controls for the X-gal staining. β-galactosidase activity was observed in three locations: in the developing lung, intestine and urinary tissue.

## MYL9 expression is ablated by the *Myl9* knockin/out allele

We wanted to confirm that the targeted *Myl9* allele was indeed ablating MYL9 expression. RNA was extracted from the bladder and small intestine of pups on the day of birth. qPCR was then performed using primers to detect exon 4 of the endogenous transcript or LacZ in the mutant transcript. *Myl9* mRNA was clearly absence from the bladder (Fig 3A) and small intestine of (S1 Fig) of *Myl9*-/- pups, while *LacZ* mRNA was expressed in its place.

Next, we performed immunohistochemical staining of these tissues with an antibody against MYL9/12A/12B. The >96% amino acid homology between these three proteins means

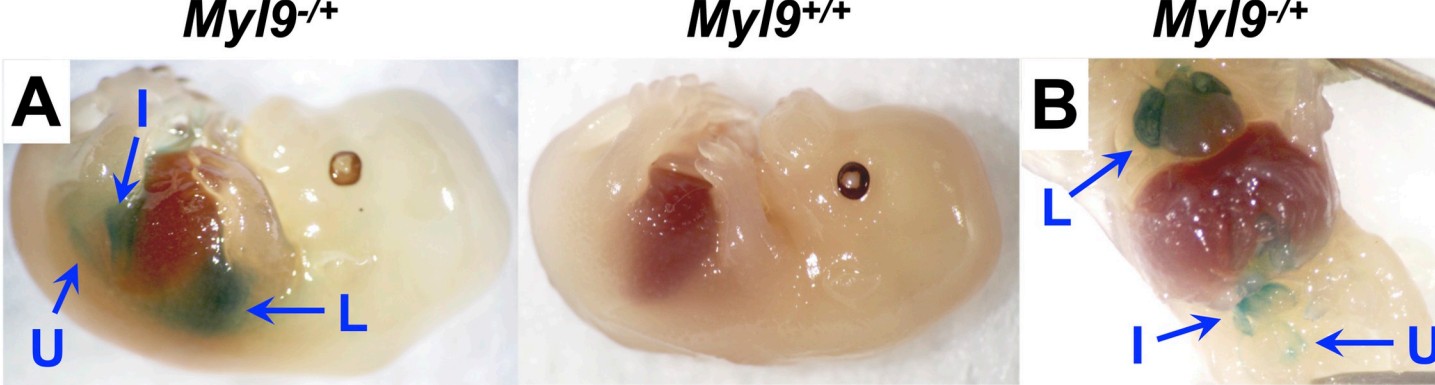

**Fig 2. *Myl9* promoter activity in the developing lung, intestine and urinary during embryogenesis. (A)** Heterozygous *Myl9*-/+ female mice were time-mated with wildtype *Myl9*+/+ male mice, then embryos were harvested at gestational day E14.5. Whole embryo X-gal staining was performed to detect β-galactosidase activity as a reporter for the *Myl9* gene. A representative *Myl9*-/+ (LacZ reporter) and *Myl9*+/+ (control) pair of embryos is shown. **(B)** The thoracic and abdominal cavities of a *Myl9*-/+ embryo was exposed following whole embryo X-gal staining. L = lung; I = intestine; U = urinary tissue.

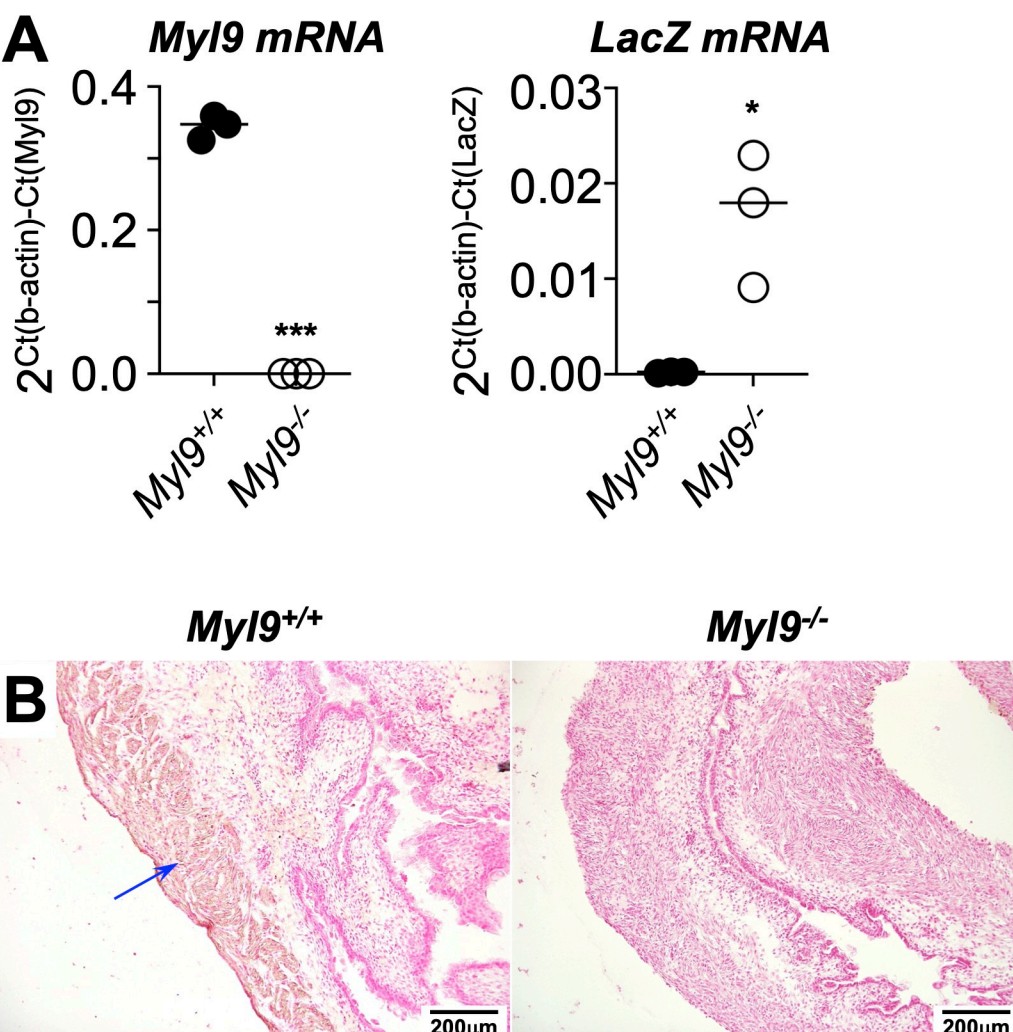

**Fig 3. Confirmation that the *Myl9* knockin/out allele ablates MYL9 expression. (A)** RNA from the bladder of *Myl9*$^{+/+}$ and *Myl9*$^{-/-}$ pups at D0 post-partum were analyzed for the expression of *Myl9* (wildtype) or *LacZ* mRNA. Each circle is an individual animal, with means indicated by the lines. Comparisons were assessed by t-test (* $P < 0.05$, *** $P < 0.0005$). The individual data points are provided in S2 File. **(B)** Sections of the bladder of a representative pair of pups were stained with an anti-MYL9/12A/12B antibody (brown stain) and counterstained with nuclear fast red. The blue arrow indicates staining in the muscularis propria of the *Myl9*$^{+/+}$ bladder.

that antibodies are unable to differentiate between them. A comparison of tissue mRNA expression of the three genes in the Broad Institute's GTEx Portal (https://gtexportal.org/home/) suggests that the bladder and intestine express only low levels of MYL12B and little MYL12A (S2 Fig). Thus, any staining by the anti-MYL9/12A/12B antibody in these tissues is mostly likely detecting MYL9 rather than the other two proteins. Indeed, while antibody staining was detected in bladder and small intestine of *Myl9*$^{+/+}$ pups, little staining was detected in *Myl9*$^{-/-}$ tissues (Figs 3B and S1).

## Bladder distension in newborn MYL9 deficient pups

Given the X-gal staining in the lung, intestine and urinary tissue of *Myl9*$^{+/-}$ embryos, as well as *Myl9* RNA and protein expression, we therefore examined these organs of pups on the day of birth for any abnormalities. Exposure of the abdominal cavity revealed that the bladder of

every $Myl9^{-/-}$ pup was abnormally enlarged with a buildup of urine (Fig 4A). Extensive structural abnormalities could be seen in histological sections of the bladder. The outer muscularis propria layer of the bladder of $Myl9^{-/-}$ pups appeared hypertrophic and disordered (Fig 4B). This is unlike the bladder of $Myl9^{+/+}$ pups, which have a uniform muscularis layer. The lamina propria of the $Myl9^{-/-}$ bladder was also severely disrupted, which exhibited thinning and a lack of rugae. These structural abnormalities suggests that $Myl9^{-/-}$ pups are unable to expel urine, thus resulting in bladder distension. This severe bladder distension could underlie the neonatal lethality observed.

Because MYL9 was expressed in the developing urinary tissue, we also examined the kidney but found no obvious abnormalities (not shown).

## Newborn MYL9 deficient pups display shortened intestines

We next examined the small intestine. The small intestine of $Myl9^{-/-}$ pups was found to be substantially shorter than that of $Myl9^{+/+}$ pups (Fig 4C). Structural abnormalities were also evident in histological sections (Fig 4D). Analogous to the bladder, the outer muscularis propria of the small intestine of $Myl9^{-/-}$ pups was hypertrophic and disordered. The serosa also appeared irregular. Furthermore, the ordered villus structure normally present was disrupted in $Myl9^{-/-}$ animals. This disruption of the small intestine may impair adequate nutrient absorption in $Myl9^{-/-}$ pups and therefore could also underlie the neonatal lethality.

## Alveola overdistension in lungs of newborn MYL9 deficient pups

Finally, we examined the lung of newborn pups. In histological sections, overdistension of the alveoli was observed throughout the lung of $Myl9^{-/-}$ animals (Fig 4E). Each alveolus in $Myl9^{-/-}$ animals appeared much larger than in their $Myl9^{+/+}$ littermates and there appeared to be fewer alveoli. This lung abnormality could also underlie the neonatal lethality in $Myl9^{-/-}$ animals.

## MYL9 expression is localized to the smooth muscle layers of the bladder, small intestine and lung

To better understand the abnormalities observed in the bladder, small intestine and lung of $Myl9^{-/-}$ animals, we performed X-gal staining of these organ of $Myl9^{+/-}$ mice to determine which specific cells normally express MYL9.

Examination of the bladder revealed that β-galactosidase activity was localized to the muscularis propria (Fig 5A). Smooth muscle cells were identified by immunohistochemical staining for α-SMA expression. Interestingly, the inner circular muscle appeared to stain more strongly than the outer longitudinal muscle. This X-gal staining is consistent with the anti-MYL9/12A/12B antibody staining found earlier (Fig 3B).

In small intestine, β-galactosidase activity was also localized specifically to muscularis propria (Fig 5B), and like in the bladder, the inner circular muscle stained more strongly than the outer longitudinal muscle. Again, this X-gal staining appears consistent with the anti-MYL9/12A/12B antibody staining (S1B Fig).

In lung, strong β-galactosidase activity was observed surrounding the bronchi, with weaker staining also observed surrounding major blood vessels (Fig 5C). Co-staining for α-SMA revealed that it is the smooth muscle surrounding the bronchi that expresses MYL9.

## MYL9 is expressed by the smooth muscle of blood vessels

With the detection of X-gal staining in the major blood vessels of the lung, we wondered if other blood vessels express MYL9. β-galactosidase activity was also observed in the smooth

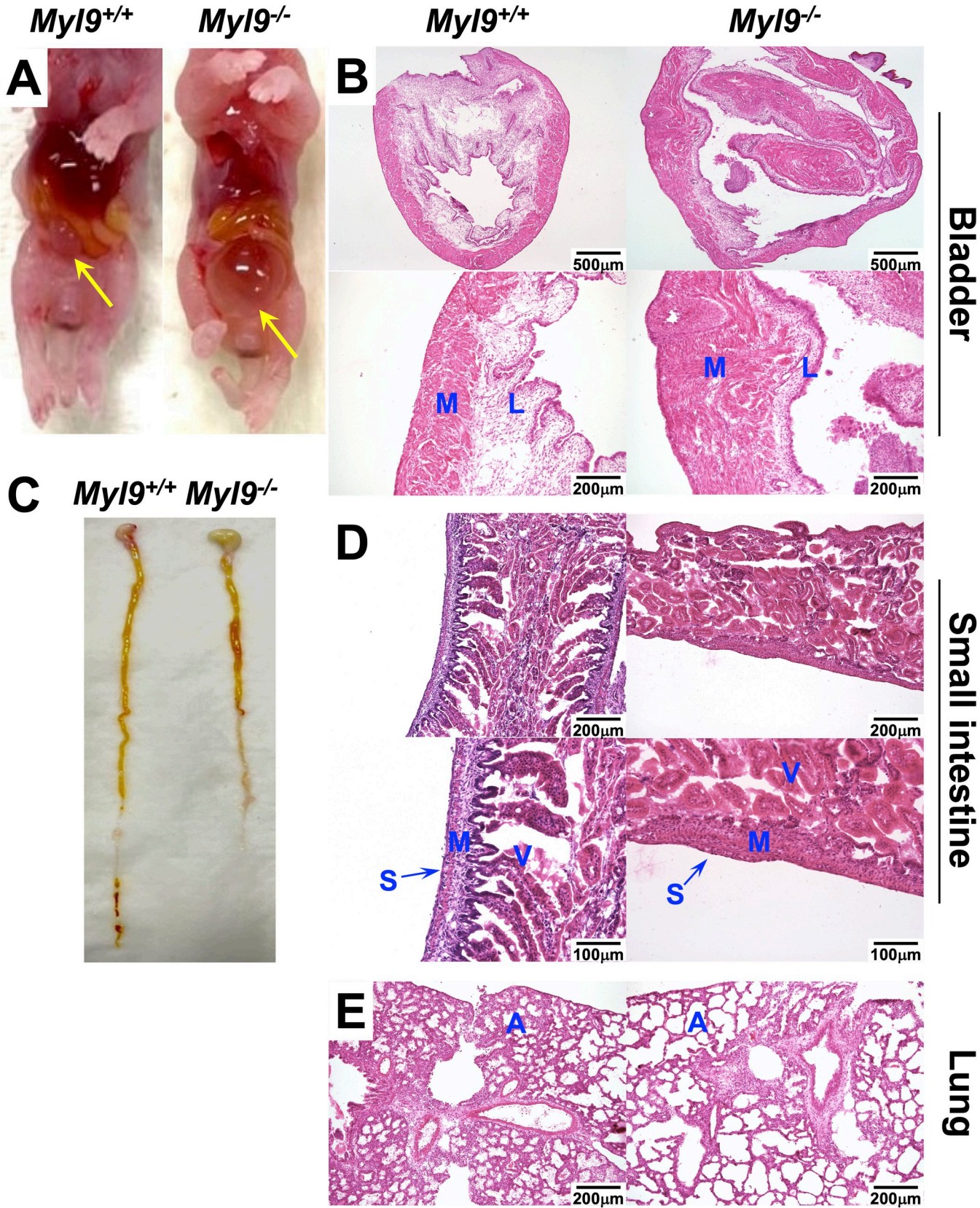

**Fig 4. Newborn MYL9 deficient pups display severe abnormalities in multiple organs.** Newborn pups were harvested within 24 hours of birth for analysis. (**A**) Myl9 deficiency causes severe bladder distention. Shown are pups with the abdominal cavity exposed, with the bladder indicated by the yellow arrow. (**B**) H&E sections of the bladder. (**C**) $Myl9^{-/-}$ pups also exhibited shortening of the small intestine. (**D**) H&E sections of the small intestine. (**E**) H&E sections of the lung. Low and high magnification images are shown for the bladder and small intestine. M = muscularis propria; L = lamina propria; S = serosa; V = villi; A = alveoli.

muscle cells of cardiac blood vessels (Fig 6A and 6B) as well as in the aorta (Fig 6C). In fact, the smooth muscle layer of all major blood vessels in all organs were positive for *Myl9* promoter activity. This included in the pancreas (Fig 6D) and kidney (Fig 6E and 6F). This suggests that MYL9 may also have a role in the smooth muscle of blood vessels.

## No evidence for significant MYL9 expression in hematopoietic cells

In our X-gal analysis of tissues from $Myl9^{-/+}$ mice, β-galactosidase activity was only ever detected in α-SMA⁺ cells and not in any non-smooth muscle cells. This was somewhat surprising given the reports of MYL9 expression in endothelial cells and various hematopoietic cells, such as T cells [8, 9, 11]. It is possible that X-gal staining may not be sensitive enough to detect low level *Myl9* promoter activity. We therefore employed a more sensitive fluorescence-activated cell sorting (FACS)-based technique to detect β-galactosidase instead. Hematopoietic cells from the spleen, thymus and bone marrow of $Myl9^{-/+}$ and $Myl9^{+/+}$ mice were analyzed. We were unable to detect β-galactosidase activity in αβ T cells, γδ T cells or B cells from the spleen (Fig 7A), despite β-galactosidase activity being clearly detected in the positive control. Similarly, no activity was detected in thymocytes (Fig 7B). We previously showed that *Myl9* mRNA is detectable at low levels in bone marrow hematopoietic stem cells [14]. Consistent with this, we detected low levels of galactosidase activity in the Lineage⁻Sca1⁺cKit⁺ (LSK) cells of the bone marrow, which contains the hematopoietic stem cell population (Fig 7C). Some expression was also detected in common myeloid progenitors but not common lymphoid progenitors (Fig 7C). Thus, under normal conditions, it appears that MYL9 is primarily expressed in the smooth muscle cells of major blood vessels, the bronchi and the muscularis propria of the small intestine and bladder. Of course, our analysis was not exhaustive and there may be other rare cells that expressed low levels of MYL9.

## Discussion

We have shown that MYL9 is an MLC that is critical for the function of smooth muscle and that deficiency is neonatal lethal in mice. This restriction just to smooth muscle was somewhat unexpected given the reports of MYL9 functioning in a diverse range of cell types and the ability to interact with a different MHC, both in muscle and non-muscle. Moreover, it was principally thought to be an important non-muscle myosin component [6]. MYL9 deficient mice was also reported recently by another group [19]. They observed the same neonatal lethality, distended bladder and shortening of the small intestine that we described. Thus, MYL9 is clearly has important physiologic functions.

Megacystis–microcolon–intestinal hypoperistalsis syndrome (MMIHS) is a severe disease characterized by impaired muscle contraction of the intestine and the bladder, resulting in hypoperistalsis and megacystis during the perinatal period [20]. This is a sporadic condition and is most frequently associated with *de novo* mutations in *ACTG2*, which encodes for γ-smooth muscle (γ-SMA) [21, 22]. Mutations in *MYH11*, which encodes for a myosin II MHC isoform, have also been shown to cause MMIHS [23]. However, two independent studies recently found an association of MMIHS with *MYL9* gene mutations. One study identified a patient with homozygous deletion of *MYL9* [24], while the other identified a patient with

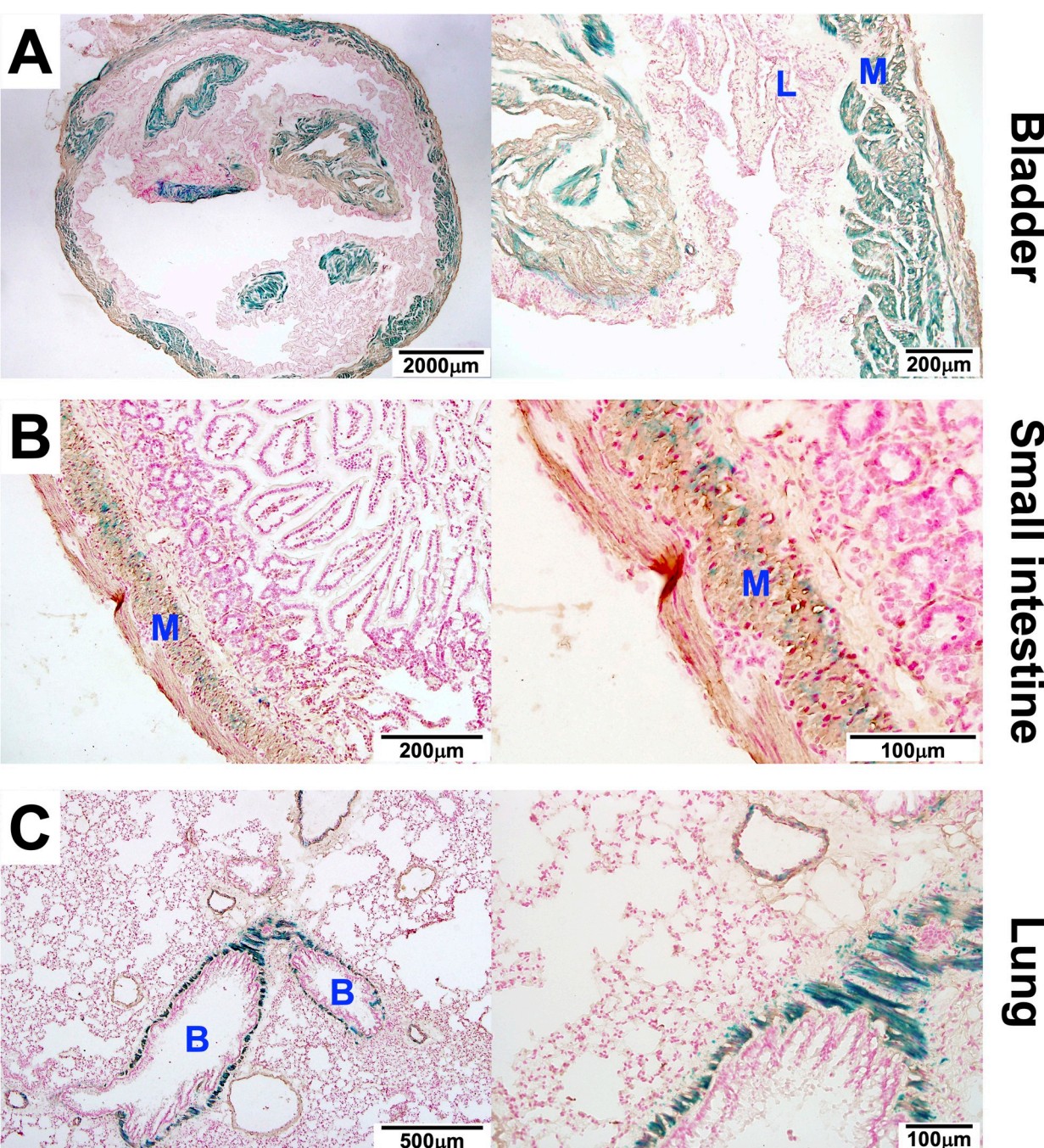

**Fig 5. MYL9 is highly expressed in the muscularis propria of the bladder and intestine, and in the bronchial smooth muscle layer.** Sections of the (**A**) bladder, (**B**) small intestine and (**C**) lung from a representative *Myl9*$^{-/+}$ mouse was stained with X-gal to detect *Myl9* promoter activity and an anti-α-SMA antibody (brown stain) to identify smooth muscle. The sections were counterstained with nuclear fast red. Low and high magnification images are shown. M = muscularis propria; L = lamina propria; S = serosa; B = bronchus.

compound heterozygous loss of function mutations [25]. Bronchopulmonary dysplasia was also observed in one of these patients [25]. These phenotypes in the intestine, bladder and lung are consistent with those seen in *Myl9*$^{-/-}$ mice. This confirms that MMIHS can indeed be caused by mutations in the *MYL9* gene. This also suggests that intestine and bladder

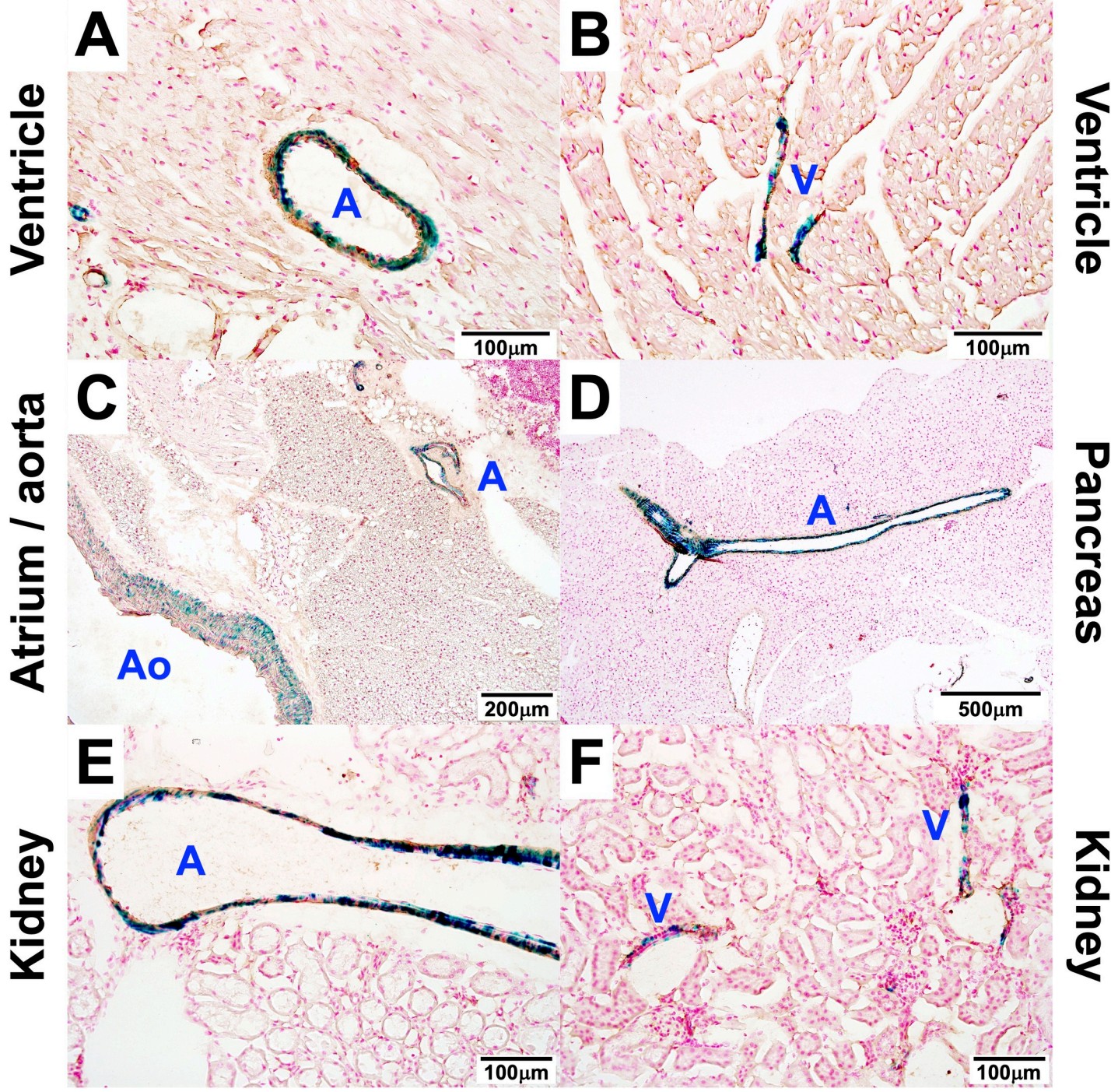

**Fig 6. MYL9 is expressed in the smooth muscle of blood vessels.** Histological sections of the **(A, B)** ventricle of the heart, © aorta and atrium of the heart, **(D)** pancreas and **(E, F)** kidney from a representative *Myl9*⁻/+ mouse were stained with X-gal to detect *Myl9* promoter activity and an anti-α-SMA antibody (brown stain) to identify smooth muscle. A = artery; Ao = aorta; V = vein.

dysfunction are the most likely cause of the neonatal lethality in *Myl9*⁻/⁻ mice. Moreover, intestinal and bladder abnormalities and neonatal lethality occurs in *Myh11*⁻/⁻ mice [26].

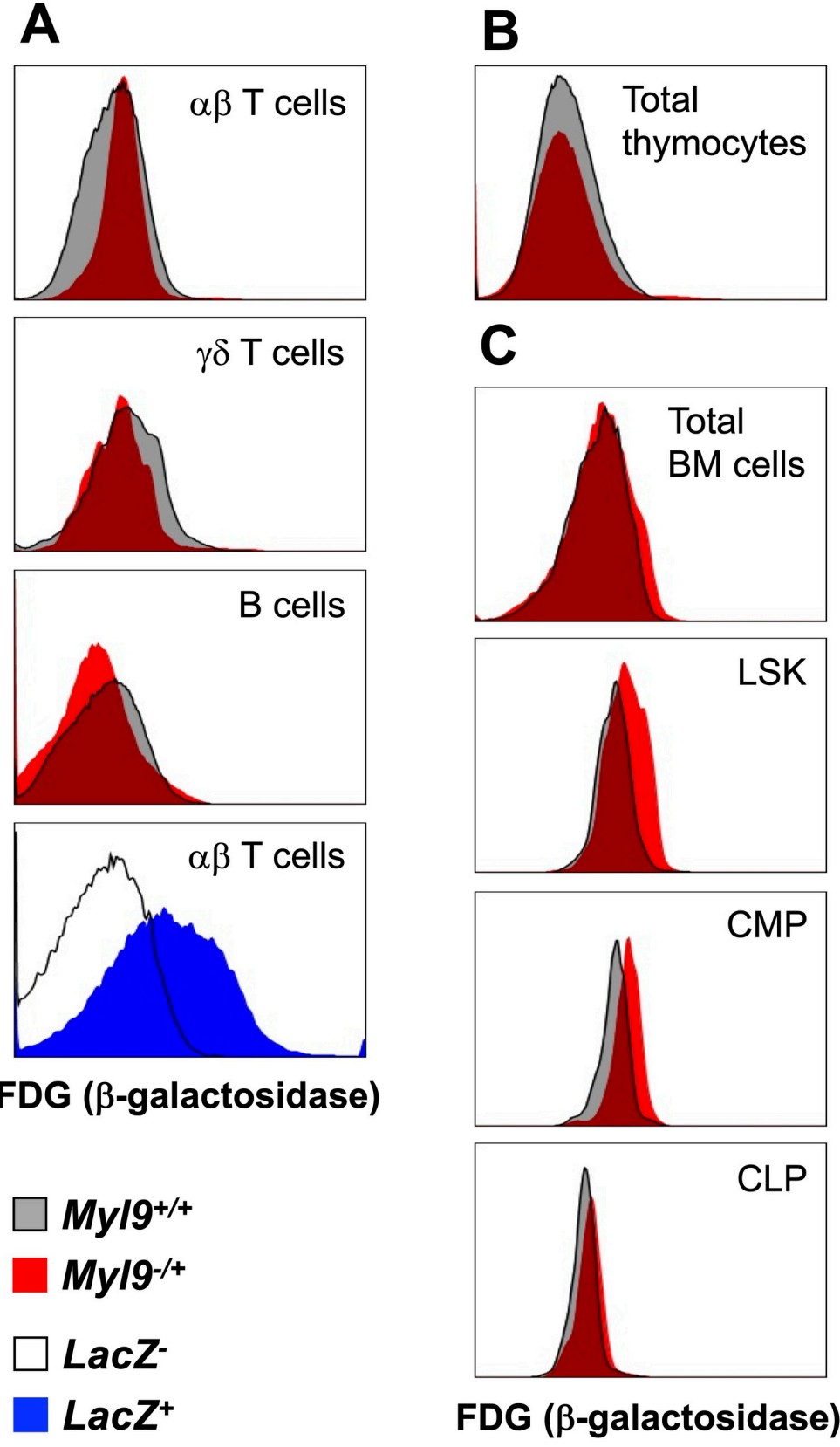

FDG (β-galactosidase)

Myl9+/+

Myl9-/+

LacZ-

LacZ+

**Fig 7.** *Myl9 promoter* **activity is undetectable in most hematopoietic cells. (A)** FACS-gal analysis of TCRβ⁺ (αβ T cells, TCRγδ T cells and B220⁺ B cells in the spleen. As a positive control, LacZ$^{Tg}$ TCRβ⁺ splenocytes were also analyzed. **(B)** Analysis of total thymocytes. **(C)** Analysis of total bone marrow (BM) cells, Lineage⁻Sca1⁺cKit⁺ (LSK) gated BM cells, which containing the hematopoietic stem cells, common myeloid progenitor (GMP) identified as Lineage⁻Sca1⁻cKit⁺CD34⁺CD16/32$^{lo}$ or common lymphoid progenitor (CLP) identified as Lineage⁻Sca1⁺cKit$^{int}$CD127⁺CD135⁺. A representative of five independent experiments is shown.

Whether MYL9 deficient MMIHS patients also exhibit vascular abnormalities are unclear. Given that we detected *MYL9* expression in major blood vessels with our LacZ reporter, MYL9 deficiency may eventually affect the vasculature of these patients. That being said, vascular smooth muscle dysfunction has yet to be reported in MMIHS patients more generally [27]. However, vascular abnormalities, including ascending aortic aneurysms, have been described in patients with multisystemic smooth muscle dysfunction syndrome (MSMDS) that is caused by mutations in the *ACTA2* gene, encoding α-SMA [28]. These patients develop a multiorgan dysfunction syndrome that overlaps with MMIHS, including dysfunction of the bladder and intestine.

The overlapping phenotypes of MYL9 deficient patients and mice with that of MYH11 deficient patients and mice, as well as ACTG2 and ACTA2 deficient patients suggests that these proteins might interact in the same pathways. γ-SMA and γ-SMA are two of the six isoforms of actin, both of which are specifically expressed by smooth muscle, although not all smooth muscle cells express both isoforms at the equal levels [29]. MYL9 is also localized to smooth muscle. Within cells, however, γ-SMA and γ-SMA have different intracellular localizations. While γ-SMA fibers are distributed throughout the cell, γ-SMA-containing fibers appear to be restricted to the center of the cell and are excluded from lamellae [29]. In this same study, the authors showed that γ-SMA is important for maintaining cell size, whereas γ-SMA is important for contractile properties.

Whether different actin isoforms specifically interact with different MHCs in smooth muscle is not entirely clear, but the interaction of MHCs in these cells may be regulated by MYL9. A previous study showed that immunoprecipitation of lysates with an antibody that recognizes MYL9, MYL12A and MYL12B pulls down different myosin II MHCs in different tissues [6]. In the bladder, MYL9/12A/12B interacts with MYH11, whereas they interact with MYH1 and MYH2 in skeletal muscle. Although this previous study was unable to distinguish between interactions with MYL9, MYL12A or MYL12B, from our study, we know that MYL9 is highly expressed in the smooth muscle of the bladder but not expressed in skeletal muscle. Thus, MYL9 quite possibly interacts with MYH11. Indeed, hollow organs from smooth muscle-specific MYL9 deficient mice exhibit impaired contraction when stimulated in vitro [19]. That being said, further analysis is required to determine precisely what MYL9 interacts with and how it functions within smooth muscle cells.

Although with our LacZ reporter we did not detect any significant MYL9 expression beyond smooth muscle, our analysis was only in normal healthy mice. MYL9 is reportedly upregulated in several cancers [12, 13]. What MYL9 is doing in cancer cells remains unclear, but it thought to promote proliferation and cell migration. MYL9 interacts with the transcriptional regulator YAP1 in colon carcinoma cells and alters the expression of YAP1-regulated genes [12]. Thus, MYL9 also has functions beyond regulating muscle contraction. Whether YAP1 is also involved in other cell types, like in hematopoietic stem cells [14], remains to be determine. Clearly, there is still much we do not understand about the functions of MYL9, both in smooth muscle and non-muscle cells.

## Supporting information

**S1 Fig. Loss of MYL9 expression in the small intestine of *Myl9*$^{-/-}$ mice. (A)** RNA from the small intestine of *Myl9*$^{+/+}$ and *Myl9*$^{-/-}$ pups at D0 post-partum were analyzed for the

expression of *Myl9* (wildtype) or *LacZ* mRNA. Each circle is an individual animal, with means indicated by the lines. Comparisons were assessed by t-test (* $P < 0.05$, ** $P < 0.005$). The individual data points are provided in S3 File. **(B)** Sections of the jejunum of the small intestine were stained with an anti-MYL9/12A/12B antibody (brown stain) and counterstained with nuclear fast red. The blue arrow indicates staining in the muscularis propria of the *Myl9*[+/+] intestine.
(TIF)

**S2 Fig. Tissue distribution of *MYL9, MYL12A* and *MYL12B* mRNA expression.** Shown is RNAseq data obtained form the Broad Institute's GTEx Portal (https://gtexportal.org/home/). Expression is shown as transcripts per million (TPM).
(TIF)

**S1 File. The individual data points for the graph in Fig 1B.** Presented is the genotyping results of individual embryos/pups at the indicated time points.
(CSV)

**S2 File. The individual data points for the graph in Fig 3A.** Presented is the relative expression of Myl9 (wildtype) mRNA versus LacZ mRNA in the bladder of individual pups analyzed by quantitative RT-PCR. The data is expressed as $2^{(Ctbeta-actin-CtMyl9)}$ or $2^{(Ctbeta-actin-CtLacZ)}$.
(CSV)

**S3 File. The individual data points for the graph in S1 Fig.** Presented is the relative expression of Myl9 (wildtype) mRNA versus LacZ mRNA in the small intestine of individual pups analyzed by quantitative RT-PCR. The data is expressed as $2^{(Ctbeta-actin-CtMyl9)}$ or $2^{(Ctbeta-actin-CtLacZ)}$.
(CSV)

## Acknowledgments

The authors thank Phenomics Australia for microinjection of embryonic stem cell clones to produce the MYL9 deficient mice and St Vincent's Hospital Melbourne's BioResources Centre's Rhiannon Walder and Lucy Kloboucnik assistance with the timed-pregnancies and expert animal husbandry.

## Author Contributions

**Conceptualization:** Chu-Han Huang, Mark M. W. Chong.

**Data curation:** Chu-Han Huang, Joyce Schuring, Jarrod P. Skinner, Lawrence Mok.

**Formal analysis:** Chu-Han Huang, Joyce Schuring, Jarrod P. Skinner, Lawrence Mok, Mark M. W. Chong.

**Funding acquisition:** Mark M. W. Chong.

**Investigation:** Chu-Han Huang, Lawrence Mok.

**Methodology:** Chu-Han Huang, Jarrod P. Skinner, Lawrence Mok.

**Supervision:** Lawrence Mok, Mark M. W. Chong.

**Visualization:** Chu-Han Huang, Lawrence Mok.

**Writing – original draft:** Chu-Han Huang, Mark M. W. Chong.

**Writing – review & editing:** Lawrence Mok, Mark M. W. Chong.

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
