## [Decision Letter · Decision Letter 0]

2 Feb 2022

PONE-D-22-00900

MYL9 deficiency is neonatal lethal in mice due to abnormalities in the lung and the muscularis propria of the bladder and intestine

PLOS ONE

Dear Dr. Chong,

Thank you for submitting your manuscript to PLOS ONE. After careful consideration, we feel that it has merit but does not fully meet PLOS ONE’s publication criteria as it currently stands. Therefore, we invite you to submit a revised version of the manuscript that addresses the points raised during the review process.

We look forward to receiving your revised manuscript.

Kind regards,

Seungil Ro, PhD

Academic Editor

PLOS ONE

Journal Requirements:

Reviewers' comments:

Reviewer's Responses to Questions

**Comments to the Author**

1. Is the manuscript technically sound, and do the data support the conclusions?

Reviewer #1: Partly

Reviewer #2: Partly

2. Has the statistical analysis been performed appropriately and rigorously? 

Reviewer #1: Yes

Reviewer #2: No

3. Have the authors made all data underlying the findings in their manuscript fully available?

Reviewer #1: Yes

Reviewer #2: Yes

4. Is the manuscript presented in an intelligible fashion and written in standard English?

Reviewer #1: Yes

Reviewer #2: Yes

5. Review Comments to the Author

Reviewer #1: Huang et al., established a line of Myl9-/- mice and characterized the phenotypes briefly. It would be interesting to explore the role of Myl9 beyond smooth muscle contraction in this animal model. However, I feel a little confusing about the logic of this paper. The aim of this paper is try to explore the in vivo physiological roles of Myl9 such as hematopoietic stem cells as author mentioned in abstract and introduction. But, the authors did not measure the corresponding functions in MYL9-/-

mice. Particularly, the hematopoietic process of the mutant mice was not described or discussed. If do not have this data, it seems there is no new finding of this report.

Major

1. P16 Line 348: the authors claim “from our study, we now know that MYL9 is highly expressed in the smooth muscle of the bladder but not expressed by skeletal muscle”. It is not true, it has been well known that MYL9 is specifically expressed in smooth muscle.

2. P15: the authors discuss too much about the overlapping phenotypes of MYL9-deficient mice and

other mutant mice as well as gene-mutated patients, because this paper did not provide any data about the interaction of these proteins.

3. Lymphocytes did not express MYL9 as claimed in this report, how about the function in MYL9-deficient lymphocytes.

Minors

1. P3 last line: have role should be have a role

2. P4 line 72: “.. is activity transcribed” means “..actively transcribed”?

Reviewer #2: The manuscript "MYL9 deficiency is neonatal lethal in mice due to abnormalities in the lung and the muscularis propria of the bladder and intestine" by Huang et al, is a promising start to understanding the nuances of different myosin isoforms. They deftly produced a Myl9 deficient mouse that also expresses LacZ in its place, thus revealing where Myl9 has been abrogated. There are several issues needing addressing before this manuscript should be considered for publication that are listed below:

1. Quantification of knockout

There is a severe lack of quantitative information about the knockout of Myl9 in their mice. They mention that genotyping was performed to show the increased LacZ in the place of exons 3 and 4 but do not attach a gel showing that this is indeed the case. This is a minor issue that can be added to supplementary data but should be attached. Even further, the authors could add supplemental sequencing data from the creation of the mouse line ordered from KOMP to increase confidence in successful insertion and removal of active Myl9 expression and genotype.

Next, considering that the authors' construct removes exons 3 & 4 of the Myl9 gene and replace it with LacZ, this presents an opportunity for quantification. While acknowledging that Myl9 shares extremely high sequence similarity with Myl12a Myl12b, designing primers within/across exons 3 & 4 of Myl9 would be ideal for qPCR quantification of Myl9 mRNA. By choosing primers containing distinct sequences within Myl9 you would likely show a decrease in Myl9 transcripts, even if there is 100% overlap of primers with Myl12a & Myl12b (as there does not appear to be any functional redundancy or compensation). Additionally, by quantifying LacZ mRNA, you could show that the loss of wild-type Myl9 has happened and that LacZ expression has taken it's place.

2. Protein Expression of MYL9

Fully understanding that Myl9 and Myl12a and Myl12b have incredibly high sequence homology, I agree that detecting Myl9 separate from Myl12a and Myl12b is extremely difficult. However, there are indeed MYL9 antibodies commercially available that could be utilized. Similar to my qPCR suggestion, the antibody might not be able to distinguish MYL9, MYL12A and MYL12B but it would show a reduction in expression within the high expressing smooth muscle cells/tissue through immunohistochemistry or western blotting. While the authors are correct that sequence similarity makes it very hard to distinguish these proteins, they do not present any data to support this other than amino acid sequence similarity. The authors could state that there was a reduction of regulatory light chains (RLC) instead of MYL9 itself. When this is combined with qPCR data showing a loss of wild-type Myl9 it would solid evidence of protein loss. Despite sequence homology, I believe data concerning protein level difference of MYL9 or RLC in general is necessary.

3. Additional FACS

The authors use FACS to isolate TCR-beta+ cells to test levels of beta-galactosidase and find a lack of expression, which goes against previous data showing that it is expressed. I feel that this needs to be resolved further with other, more general hematopoietic cell markers such as CD45 or CD34 and then isolating either RNA/protein from these cells and quantifying Myl9/LacZ expression. It is a striking conundrum that the beta-galactosidase is only showing up in smooth muscle but not other known cellular sources. This undercuts their contention that knockout is indeed globally accomplished and is in direct contrast to previous publications. At the very least, further exposition as to why this contrast is occurring is needed.

4. Poor Image Quality

The cryosection images throughout the manuscript must be of better resolution. By mentioning that hypertrophy is occurring in both the bladder and intestinal muscle layers, the authors have presented another opportunity for quantification that is not present. With clearer images, the authors will be able to show how much hypertrophy is present in both qualitative and quantitative forms. Within the intestinal images, it is unclear what level of hypertrophy is present and without a clearer image and quantification, it will remain unclear.

5. Minor English Editing

There are a few incorrectly used (abnormal instead of abdominal on pg. 9 of submission) and missing words in the manuscript but this is of minor concern.

With these additions and corrections, I believe the manuscript will be of higher quality and complete for submission.

6. PLOS authors have the option to publish the peer review history of their article (what does this mean?). If published, this will include your full peer review and any attached files.

Reviewer #1: No

Reviewer #2: **Yes: **Brian G Jorgensen

---

## [Author Response · Author response to Decision Letter 0]

18 Apr 2022

A point-by-point reply the reviewer's comments are attached as a word document.

---

## [Decision Letter · Decision Letter 1]

12 Jun 2022

PONE-D-22-00900R1MYL9 deficiency is neonatal lethal in mice due to abnormalities in the lung and the muscularis propria of the bladder and intestinePLOS ONE

Dear Dr. Chong,

Thank you for submitting your manuscript to PLOS ONE. After careful consideration, we feel that it has merit but does not fully meet PLOS ONE’s publication criteria as it currently stands. Therefore, we invite you to submit a revised version of the manuscript that addresses the points raised by the Reviewer 1.

Please submit your revised manuscript by Jul 27 2022 11:59PM If you will need more time than this to complete your revisions, please reply to this message or contact the journal office at plosone@plos.org. Please include the following items when submitting your revised manuscript:A rebuttal letter that responds to each point raised by the academic editor and reviewer(s). You should upload this letter as a separate file labeled 'Response to Reviewers'.A marked-up copy of your manuscript that highlights changes made to the original version. You should upload this as a separate file labeled 'Revised Manuscript with Track Changes'.An unmarked version of your revised paper without tracked changes. You should upload this as a separate file labeled 'Manuscript'.If applicable, we recommend that you deposit your laboratory protocols in protocols.io to enhance the reproducibility of your results. Protocols.io assigns your protocol its own identifier (DOI) so that it can be cited independently in the future. For instructions see: https://journals.plos.org/plosone/s/submission-guidelines#loc-laboratory-protocols. Additionally, PLOS ONE offers an option for publishing peer-reviewed Lab Protocol articles, which describe protocols hosted on protocols.io. Read more information on sharing protocols at https://plos.org/protocols?utm_medium=editorial-email&utm_source=authorletters&utm_campaign=protocols.

We look forward to receiving your revised manuscript.

Kind regards,

Seungil Ro, PhD

Academic Editor

PLOS ONE

Journal Requirements:

Reviewers' comments:

Reviewer's Responses to Questions

**Comments to the Author**

1. If the authors have adequately addressed your comments raised in a previous round of review and you feel that this manuscript is now acceptable for publication, you may indicate that here to bypass the “Comments to the Author” section, enter your conflict of interest statement in the “Confidential to Editor” section, and submit your "Accept" recommendation.

Reviewer #1: All comments have been addressed

Reviewer #2: All comments have been addressed

2. Is the manuscript technically sound, and do the data support the conclusions?

Reviewer #1: Yes

Reviewer #2: Yes

3. Has the statistical analysis been performed appropriately and rigorously? 

Reviewer #1: Yes

Reviewer #2: Yes

4. Have the authors made all data underlying the findings in their manuscript fully available?

Reviewer #1: Yes

Reviewer #2: Yes

5. Is the manuscript presented in an intelligible fashion and written in standard English?

Reviewer #1: Yes

Reviewer #2: Yes

6. Review Comments to the Author

Reviewer #1: The authors clarified the purpose of this study is investigating the physiological function of Mly9 because the physiological function of Mly9 remains unclear. The result shows the importance of Myl9 in smooth muscle. It has been documented that Myl9 serves as the regulatory light chain (RLC) of smooth muscle myosin and initiates smooth muscle contraction when it is phosphorylated by Mylk. The function of Myl9 is clear at least in smooth muscle. Aso, there are reports showing that tissue specific or global deletion of Myl9 causes abolished smooth muscle contraction and hence the dilation phenotypes of hollow organs. The question is what is the new finding of this study?

Reviewer #2: Huang et al, have properly addressed each issue that I raised in my initial review and gone above and beyond to resolve the questions and concerns that were initially expressed. The addition of the use of the Myl9/Myl12a/Myl12b antibodies, with LacZ protein data, adds a strong component to the paper allowing for direct visualization of clear loss of the protein in the smooth muscle layer. Furthermore, the completion of more robust FACS information via the addition of other hematopoietic markers to solidify their conclusions on isolated cells. The combination of the molecular and gross anatomical data that Huang et al have responded to my comments with was thoughtful, well-crafted, and much more complete for publication at this time.

7. PLOS authors have the option to publish the peer review history of their article (what does this mean?). If published, this will include your full peer review and any attached files.

Reviewer #1: No

Reviewer #2: **Yes: **Brian Jorgensen

---

## [Author Response · Author response to Decision Letter 1]

13 Jun 2022

It has been documented that Myl9 serves as the regulatory light chain (RLC) of smooth muscle myosin and initiates smooth muscle contraction when it is phosphorylated by Mylk. The function of Myl9 is clear at least in smooth muscle. Also, there are reports showing that tissue specific or global deletion of Myl9 causes abolished smooth muscle contraction and hence the dilation phenotypes of hollow organs. The question is what is the new finding of this study?

Although not stated by the reviewer, we believe that they are referring to the studies from Sun et al (Front Physiol, 11:593966) and Park et al (Biochem J, 434:171). These are two previous studies that are discuss in our manuscript (4th paragraph of introduction and multiple locations in the discussion sections). As we discussed, Park et al.’s study suggests that MYL9 is important for non-muscle myosin. Sun et al’s study does indeed show that the intestine and bladder from smooth muscle-specific (cre/LoxP) MYL9 deficient mice exhibit impaired contraction in vitro. However, these mice had a mild phenotype compared to germline MYL9 deficient mice. Thus, it is unclear whether is due to incomplete conditional Myl9 gene deletion or if MYL9 is expressed more broadly than just smooth muscle. In fact, neither the Park nor Sun studies provide any evidence that MYL9 is actually restricted to smooth muscle.

As highlighted by the comments of Reviewer 2 (of the initial manuscript), this has been an ongoing issue for the field because of the lack of reagents that can distinguish between MYL9, MYL12B and MYL12B protein. RNA analysis, such as by RNAseq or qPCR, can distinguish between the different transcripts. However, as explained in our response to Reviewer’s 2 comments, previous studies have only been able to measure MYL9 expression in whole organs and have not been able to determine the actual cell types that expresse MYL9. Thus, it has remained unclear whether MYL9 expression is specific to smooth muscle or has a broad distribution, especially since MYL9 is detectable in almost all organs. This includes the data presented in both the Park et al and Sun et al papers referenced above.

Our study definitely shows for the first time that MYL9 expression is restricted to the smooth muscle layer of contracting organs and of bronchi and smooth muscle. Therefore, failure of smooth muscle cell function is likely the cause of the lethality in MYL9 deficient mice and MMIHS patients. To emphasize this point we have modified the abstract to state “…Using this reporter, we show that MYL9 expression is restricted to the muscularis propria of the small intestine and bladder, as well as in the smooth muscle layer of the bronchi in the lung and major bladder vessels in all organs…”

As we acknowledge the last paragraph of the discussion section, this restricted expression may only be in healthy mice as there is evidence that MYL9 is upregulated in a range of disease settings.

---

## [Decision Letter · Decision Letter 2]

21 Jun 2022

MYL9 deficiency is neonatal lethal in mice due to abnormalities in the lung and the muscularis propria of the bladder and intestine

PONE-D-22-00900R2

Dear Dr. Chong,

We’re pleased to inform you that your manuscript has been judged scientifically suitable for publication and will be formally accepted for publication once it meets all outstanding technical requirements.

Kind regards,

Seungil Ro, PhD

Academic Editor

PLOS ONE

Additional Editor Comments (optional):

Reviewers' comments:

Reviewer's Responses to Questions

**Comments to the Author**

1. If the authors have adequately addressed your comments raised in a previous round of review and you feel that this manuscript is now acceptable for publication, you may indicate that here to bypass the “Comments to the Author” section, enter your conflict of interest statement in the “Confidential to Editor” section, and submit your "Accept" recommendation.

Reviewer #1: All comments have been addressed

2. Is the manuscript technically sound, and do the data support the conclusions?

Reviewer #1: Yes

3. Has the statistical analysis been performed appropriately and rigorously? 

Reviewer #1: Yes

4. Have the authors made all data underlying the findings in their manuscript fully available?

Reviewer #1: Yes

5. Is the manuscript presented in an intelligible fashion and written in standard English?

Reviewer #1: Yes

6. Review Comments to the Author

Reviewer #1: I have no more comments and recommend this paper acceptable.

7. PLOS authors have the option to publish the peer review history of their article (what does this mean?). If published, this will include your full peer review and any attached files.

Reviewer #1: No

---

## [Editor Report · Acceptance letter]

30 Jun 2022

PONE-D-22-00900R2 

MYL9 deficiency is neonatal lethal in mice due to abnormalities in the lung and the muscularis propria of the bladder and intestine 

Dear Dr. Chong:

I'm pleased to inform you that your manuscript has been deemed suitable for publication in PLOS ONE. Congratulations! Your manuscript is now with our production department. 

Kind regards, 

on behalf of

Dr. Seungil Ro 

Academic Editor

PLOS ONE